# Symptomatic Patients with Hyperleukocytic FLT3-ITD Mutated Acute Myeloid Leukemia Might Benefit from Leukapheresis

**DOI:** 10.3390/cancers16010058

**Published:** 2023-12-21

**Authors:** Kiavasch Mohammad Nejad Farid, Tim Sauer, Michael Schmitt, Carsten Müller-Tidow, Anita Schmitt

**Affiliations:** 1Department of Internal Medicine V, University Hospital Heidelberg, 69120 Heidelberg, Germany; 2German Cancer Research Center (DKFZ), National Center for Tumor Diseases (NCT), German Cancer Consortium (DKTK), 69120 Heidelberg, Germany

**Keywords:** acute myeloid leukemia, leukapheresis, leukostasis, hyperleukocytosis, cytoreduction, early mortality

## Abstract

**Simple Summary:**

Leukostasis in hyperleukocytotic acute myeloid leukemia (AML) is a medical emergency. The optimal treatment strategy and the potential benefits of incorporating emergency leukapheresis (LA) are still unclear. In our retrospective study, we observed a striking difference in the outcome of FLT3-ITD mutated (mut) AML patients undergoing LA in the early phase of treatment compared to patients with FLT3 wildtype (wt). Knowledge of mutational status might guide treatment decisions in the early phase of hyperleukocytosis.

**Abstract:**

Purpose: We aimed to identify subsets of patients who benefit from emergency LA and to establish a therapeutic algorithm for AML patients with hyperleukocytosis. Methods: In this single-center retrospective cohort study, a total of 20 consecutive patients underwent LA because of their clinical symptoms. Overall survival (OS) analysis was conducted using the Kaplan–Meier plot method. Univariate and multivariate analyses (using multiple logistic regression) were performed. At the time of diagnosis, all patients received a standard diagnostic workup for AML including FLT3-ITD mutational analysis. Results: FLT3-ITD mut AML patients receiving LA had a median OS of 437 days (range 5–2379 days) with a corresponding 14-day survival of 92.3%, while FLT3 wt AML patients displayed a significantly lower median OS of only 5 days (range 1–203 days) with a corresponding 14-day survival of 14.3% (*p* = 0.0001). Conclusions: Among patients with clinical symptoms of leukostasis, the subset of FLT3-ITD mut AML patients showed a better outcome with lower early mortality after emergency LA. Based on these observations, we established a therapeutic algorithm for AML patients with hyperleukocytosis.

## 1. Introduction

Despite huge advances in the understanding of acute myeloid leukemia (AML) biology and in the development of novel therapies, the prognosis of the disease remains dismal, with an age-dependent 5-year survival of only 15–35% among patients [1].

Early mortality is also determined by unique complications, such as leukostasis in the setting of hyperleukocytosis (occurring in approx. 10% of all AML patients [2]), defined as a peripheral blood leukocyte count higher than 100,000/µL. Leukostasis leads to increased blood viscosity and obstruction of capillary beds, which further leads to tissue hypoxia with consecutive organ failure and eventually death. Therefore, leukostasis is considered to be a medical emergency that requires immediate intervention. Besides rapid cytoreduction by hydroxyurea and/or cytarabine, leukapheresis (LA) is a well-established method for rapid leukocytoreduction, but its benefits or effectiveness still remain controversial in the literature (see Table 1). Therefore, there is an urgent need to optimize the treatment of hyperleukocytosis. LA grants the possibility of very rapid ‘debulking’ of the leukemic tumor burden and is usually well-tolerated. Although LA lowers leukocyte counts much faster than cytoreductive chemotherapy, its effect is usually very short-lasting and requires concomitant systemic therapy to reduce blast production and replenishment into the peripheral blood. There are also several limitations to the widespread use of LA, such as its availability, relatively high cost of the procedure and the need for well-trained personnel.

AML with hyperleukocytosis displays unique genomic features. The most common molecular genetic abnormalities in cytogenetically normal AML are *Fms related tyrosine kinase receptor 3* (FLT3), *nucleophosmin 1* (NPM1) and *tet methylcytosine dioxygenase 2* (TET2) mutations [3]. Cytogenetic abnormalities associated with hyperleukocytosis are rearrangements involving the 11q23 locus (*mixed-lineage-leukemia* or *MLL* gene) or inv(16) [4]. Although not considered a significant risk factor by the European LeukemiaNet (ELN), and therefore not included in their risk classification, hyperleukocytosis seems to harbor an independent negative effect on outcomes in de novo AML [5].

**Table 1 cancers-16-00058-t001:** Literature review of leukapheresis trials in AML patients.

Author/Year	Early Mortality	Overall Survival	CR Rate	No. of Patients
Cuttner et al., 1983 [6]	-	↔	↑	LA: 22
Porcu et al., 1997 [7]	27%	↔	-	LA: 48
Thiebaut et al., 2000 [8]	*comparable to today*	-	-	LA: 53
Giles et al., 2001 [9]	↓	↔ (↓)	(↑)	Total: 146; LA: 71
Tan et al., 2005 [10]	57%	-	-	LA: 14
Chang et al., 2007 [11]	↔	(↓)	-	LA: 75
Bug et al., 2007 [12]	↓	↔	↔	Total: 53; LA: 23
De Santis et al., 2011 [13]	47%	-	-	LA: 15
Pastore et al., 2014 [2]	↔	↔	↔	Total: 52; LA: 20
Berber et al., 2015 [14]	↔	↔	-	LA: 31
Kuo et al., 2015 [15]	↔ (*if symptomatic* ↑)	↔	-	Total: 88; LA: 41
Malkan and Ozcebe, 2017 [16]	↔ (↑)	-	-	Total: 28; LA: 10
Nan et al., 2017 [17]	↓↓	↔	-	Total: 52; LA: 26
Choi et al., 2018 [18]	↔	↔	-	Total: 44; LA: 22
Rinaldi et al., 2021 [19]	↔ (40%)	-	-	LA: 38
Blum and Porcu, 2007 [20] *Review*	*-divergent results-*			LA: 141
Ganzel et al., 2012 [21] *Review*	*divergent* *results*	↔	-	Total: 386; LA: 246
Ali et al., 2016 [22] *Review*	*divergent* *results*	↔	-	-
Shallis et al., 2020 [23] *Review*	-	↔	-	Total: 219; LA: 32
Stahl et al., 2020 [24] *Review*	↔ (↓)	↔ (↑)	↔	Total: 779; LA: 113
Rinaldi et al., 2022 [25] *Review*	↔	↔	-	Total: 1407; LA: 317
*Summary of 21 trials:*	↔ (↓)	↔	(↔)	

LA: leukapheresis, CR: complete remission, ↑: increased, ↓: decreased, or ↔: (mostly) unchanged/comparable (e.g., mortality) to non-LA-treated patients (usually non-significant differences). If in parenthesis, no statistical significance but rather a tendency was observed. Note that some of the listed review articles include results from the aforementioned studies.

The purpose of this study was to investigate the role of LA in the initial treatment of AML patients with leukostasis in hyperleukocytosis. Furthermore, we aimed to identify subsets of patients who could benefit from emergency LA and to establish a therapeutic algorithm for AML patients presenting with hyperleukocytosis and symptomatic leukostasis.

## 2. Methods

### 2.1. Trial Population and Design

The study cohort consisted of twenty consecutive patients with hyperleukocytic AML who were treated with LA at the Heidelberg University Hospital between July 2014 and October 2022, as well as ten consecutive patients (September 2015–October 2022) with hyperleukocytosis treated without LA. The decision to perform LA was driven by the presence of symptomatic disease. The primary endpoints were early mortality at 14 days, one-year survival and overall survival. All data were obtained by electronic chart review. Written informed consent according to the Declaration of Helsinki for data analysis and for the leukapheresis procedure was obtained from all patients who were able to consent at the time of diagnosis. Exception from informed consent applied to unconscious/intubated patients. The local ethics committee approved the data collection and analysis.

### 2.2. Genetic Analysis

As a standard procedure, all patient samples obtained at the time of diagnosis were subjected to a cytogenetic and molecular genetic analysis by means of Giemsa stain-based karyotyping, fluorescence in situ hybridization and polymerase chain reaction testing.

### 2.3. Leukapheresis Procedure

Cytoreductive leukapheresis was performed via central venous or peripheral venous catheter(s) using Spectra Optia^®^ devices (Terumo, Tokyo, Japan). Two to four times of a patient’s total blood volume was processed during a single leukapheresis session with a leukapheresis duration of three to five hours. During the procedure, acid citrate dextrose solution A (ACD-A) was used as an anticoagulant at a ratio blood to anticoagulant of 12–15:1.

### 2.4. Statistical Methods

Survival curves for overall survival were estimated using the Kaplan–Meier method and compared between groups using the log-rank test. Categorical variables were described by absolute and relative frequencies. For continuous variables, the Student’s t-test was applied. Survival times calculations and the aforementioned tests were performed using GraphPad Prism software (release 9.3.1; San Diego, CA, USA). SPSS was used for univariate analysis (using Fisher’s exact test) and multivariate analysis using multiple logistic regression with 95%-confidence intervals. Significance levels were set at 0.05. Data were analyzed as of 15 July 2023.

## 3. Results

### 3.1. Cytoreductive Performance of Leukapheresis and Chemotherapy

Overall, 42 leukapheresis procedures were performed on 20 patients in this single-center, retrospective cohort study. The average number of LA sessions per patient was 2.1, with five of the twenty patients receiving only one LA, eight patients receiving two LAs, and seven patients receiving three LAs. The average WBC count before and after the first apheresis was 236.2/nL (range, 115.3–586.1/nL; standard deviation 109.4/nL) and 139.9/nL (range, 69.2–472.1/nL; standard deviation 91.8/nL), respectively, resulting in an average reduction in WBC count of 43.1 percent (range, 12.1–62.2) achieved by the first LA procedure alone. The average WBC count after the last LA session was 92.1/nL (range, 47.5–288.7/nL), resulting in an average total reduction in WBC count of 57.8 percent (range, 27.2–92.4). LA proved to be safe; there were no severe adverse events, fatalities, or complications of the indwelling catheters associated with the LA procedure.

Additionally, data of patients with hyperleukocytosis treated without LA were juxtaposed with the group of patients treated with LA. The average WBC count before and 48 h after initiation of treatment was 181.1/nL (range, 104.6–409.0/nL; standard deviation 90.3/L) and 132.6/nL (range, 80.9–287.9/nL; standard deviation 70.5/nL), respectively, resulting in an average reduction in WBC count of 24.4 percent (range, −24.8–63.9). While WBC count was comparable in both groups, the relative leukocyte count reduction was significantly higher in the LA group (*p* = 0.0096), indicating a very rapid and effective means of cytoreduction.

### 3.2. Patient Characteristics

Detailed individual and summarized patient characteristics are given in Table 2 and Table 3 (as well as in Table 4 and Table 5). Both of the differently treated cohorts were balanced with respect to age (LA: mean 58 years, range 18–80 years; non-LA: mean 58 years, range 30–80 years) and leukocyte counts (see above) with a male/female ratio of 1:1. Non-favorable risk AML according to the International Consensus Classification of myeloid neoplasms and acute leukemia 2022 [26] was present in 85% of patients in the LA cohort and in 70% of patients in the non-LA cohort. FLT3-ITD mutations were more commonly found in the LA-treated patients (13/20; 65% vs. 3/10; 30% in the non-LA cohort), whereas FLT3-TKD was only found in one patient in the LA cohort. Cytogenetic and molecular genetic aberrations are listed in Table 2 and Table 4. All patients in the LA cohort had symptomatic disease with probable pulmonary or cerebral leukostasis contributing to respiratory (60% of symptomatic patients) or neurological compromise (50% of symptomatic patients), while in the non-leukapheresis cohort, only 3/10 patients had clinical symptoms (10% dyspnea, 20% respiratory failure). All patients in both cohorts received cytoreductive treatment, most with hydroxyurea (HU) or cytarabine, or both.

### 3.3. Outcome after Cytoreduction/Leukapheresis

The median OS in the LA group was 194 days (range 1–2379 days; see Figure 1). The total 14-day survival and total 1-year survival were 65% and 40%, respectively. The total 14-day survival in the non-LA group was 90%, with a median OS of 412 days (range 11–2865 days), and the 1-year survival was 60%. Direct comparison of OS and early mortality between the LA and non-LA groups is not possible due to 1:1 matched pairing of patients not being applicable.

### 3.4. Prognostic Factors for Early Death: FLT3 Mutational Status

We identified FLT3 wt status to be predictive of early death in AML with hyperleukocytosis. FLT3-ITD mut AML patients treated with LA had a median OS of 437 days (range 5–2379 days) with a corresponding 14-day survival of 92.3%, while FLT3 wt AML patients displayed a significantly lower median OS of only 5 days (range 1–203 days) with a corresponding 14-day survival of 14.3% (log-rank test of OS data; *p* = 0.0001; see Figure 2). WBC counts in both FLT3-ITD mut and wt patients were comparable in both cohorts (LA or non-LA) before treatment initiation. Differences in outcome depending on FLT3 status are not explained or confounded by different WBC counts. Endothelial activation and stress index (EASIX) scores reflecting organ damage (especially renal function) and microcirculation compromise pre-treatment did not differ between the LA and non-LA cohorts or between the FLT3 mut/wt subgroups of both cohorts, despite showing a tendency to higher score values in the LA cohort (57.2 vs. 27.5; *p* = 0.1390). This is most likely explained by these patients having symptomatic disease with active/impending leukostasis, leading to the decision to perform LA (see discussion). The total 1-year survival in the LA-treated group was 61.5% in FLT3 mut AML patients, whereas none of the FLT3 wt AML patients presenting with hyperleukocytosis receiving LA survived up to one year after initial diagnosis.

### 3.5. Univariate and Multivariate Analysis of 14-Day Survival

Fisher’s exact test was performed as univariate analysis regarding possible associations between age (≥60 vs. <60 years), FLT3 mutational status, relative leukocyte reduction achieved by the first LA session (≥50% vs. <50%) and 14-day survival. In patients with hyperleukocytic AML, the presence of a FLT3-ITD mutation is associated with a significantly different 14-day survival (*p* = 0.007), whereas age and relative leukocyte reduction are not associated with any differences in early mortality.

Subsequently, multiple logistic regression as multivariate analysis was performed to identify prognostic markers of early mortality displayed by 14-day survival rates (Table 6). In patients with hyperleukocytic AML, the presence of a FLT3-ITD mutation (Odds ratio 19.02, 95% CI 1.48–682.6; *p* = 0.0424) confers a significantly superior 14-day survival. No association was found for other independent variables, such as age or the relative leukocyte reduction achieved by the first LA session.

## 4. Discussion

Mortality in AML patients can be separated into two phases: early and late mortality. Late mortality is determined by the natural biological course of the disease and is mostly due to relapse or transplant-associated complications, e.g., chronic graft-versus-host-disease. Early mortality is usually due to infectious complications, hemorrhage or (multi-)organ failure (due to leukostasis, among other causes). As mentioned previously, despite many advances in supportive care, early mortality has remained unchanged over the last few decades. Newer approaches to counteract damage induced by leukostasis include targeting the blast adhesion itself rather than solely reducing the peripheral WBC count [4] or the use of anti-inflammatory agents, e.g., dexamethasone [27].

The literature on LA is summarized in Table 1. As shown in the studies by Kuo et al. [15,28], the survival curves of patients with hyperleukocytosis are shifted to a certain delta at a very early timepoint [28]. After these early lethal events, the survival curve run in parallel to that of patients without hyperleukocytosis, with similar dynamics and plateaus. Hyperleukocytosis and its treatment (regardless of the method of cytoreduction) only seem to prognostically influence the early phase of the disease course. Beyond the first weeks, there seems to be no impact on long-term outcomes. If a patient survives the initial critical phase, the further course is dictated by the natural biological behavior of the disease and its genetic features.

The decision to perform an emergency LA was usually driven by the presence of leukostasis symptoms. Since the occurrence and severity of symptoms can vary widely, the clinical diagnosis of leukostasis was not established with a uniform algorithm but in a rather individual manner. There are several pitfalls in establishing the diagnosis of (especially pulmonary) leukostasis due to the broad differential diagnosis of respiratory failure, e.g., pneumonia, sepsis or fluid overload. Leukostasis, however, in most cases remains a clinical diagnosis (of suspicion or exclusion). In our patients, LA was not performed prophylactically or solely on the basis of a high WBC count, since there is no strong evidence for prophylactic LA. Attempts to develop a straight-forward algorithm have been made in the past [29], but usually lack objective parameters to use for decision guidance. Establishing such an algorithm is especially difficult due to the rarity of this condition and the lack of larger study cohorts. To fully grasp the real benefit-to-harm ratio of a procedure like LA in this case, other factors, i.e., cost-effectiveness and LA-related risks/complications, have to be taken into account in each individual case. All limitations associated with the nature of a retrospective cohort study should be taken into consideration when interpreting the results. Furthermore, we only had a limited sample size of 20 patients undergoing LA, which may not be adequate for analyzing all confounders. On the other hand, given the rarity of the studied clinical scenario, our sample size is comparably one of the larger cohorts in the literature, therefore contributing new and valuable information in this difficult-to-study population.

All patients treated with LA had symptoms of leukostasis. This is in contrast to the chemotherapy-only group, in which approximately one-third of patients had symptoms that could be attributed to leukostasis, making the patients in the LA group automatically more morbid and prone to further complications (related to more severe deterioration of organ function). This is reflected by the higher EASIX scores [30] of patients treated with LA (Table 2 and Table 3) when compared to the non-LA group (Table 4 and Table 5). From all patients undergoing LA, almost only the patients with FLT3-ITD mut AML survived the first 14 days after diagnosis. Regarding the striking survival difference in the early phase, one might hypothesize that FLT3-ITD mut hyperleukocytosis has a more ‘benign’ course. Alternatively, FLT3-ITD mut AML patients might benefit more from LA than FLT3 wt AML patients (Figure 2). In both cohorts, FLT3 wt patients showed a trend towards higher EASIX score values. Again, in this case, it is quite difficult to differentiate the main contributing factors causing the EASIX scores. Whether the FLT3 wt blasts intrinsically cause more endothelial damage, leading to a more ‘malignant’ form of leukostasis, or whether the FLT3 wt AML patients had more endothelial dysfunction due to other factors (infection, chemotherapy toxicity) cannot be fully deciphered.

There are several studies (e.g., Cuttner et al., 1980 [31]) highlighting the relevance of a (myelo-)monocytic differentiation when it comes to hyperleukocytosis and leukostasis syndrome. (Myelo-)monocytic morphology is thought to result in more adhesive blasts, compromising the microcirculation in vital tissues. In this study, unfortunately, the specific morphological features (or formerly FAB classification) were documented only in a minority of cases (due to the steadily decreasing significance of the morphological classification of AML in clinical practice), and therefore could not be analyzed as an independent parameter.

Blast biology has a significant impact on leukostasis severity. The expression of surface markers such as *cluster of differentiation 11c* (CD11c) on leukemic blasts is associated with a high risk of early death in AML patients with hyperleukocytosis [12]. CD11c is a member of the integrin family mediating cell adhesion, which may contribute to more tissue damage. This observation leads to the hypothesis that the extremely heterogeneous types of AML regarding long-term prognosis also differ in their initial presentation and biological behavior in the short-term early treatment period. Certain driver mutations can influence the proliferation activity and are more likely to induce a hyperproliferative state more than others. For example, core-binding-factor leukemia, defined by the recurrent cytogenetic abnormality inv(16), also frequently presents with hyperleukocytosis, but displays a much lower early mortality rate and a 5-year OS of approx. 80%. Despite being considered high-risk regarding early mortality [32], there is almost no difference in OS between patients presenting with and without hyperleukocytosis. Considering the CR rates of up to 93%, the induction mortality risk of 5% and approx. 25% of inv(16)-mut AML patients presenting with hyperleukocytosis [33], and assuming that all induction deaths happen in patients with hyperleukocytosis, the maximum early mortality is approx. 20% in patients with hyperleukocytosis. In contrast, in the literature as well as in our study, early mortality in patients with hyperleukocytosis reaches almost 40%. This is in line with the assumption that not only absolute WBC counts but also differences in genetic features could influence the harmfulness of hyperleukocytosis.

To our knowledge, this is the first paper studying the potential impact of LA depending on molecular genetic features. Our observations are in line with prior studies, in which FLT3-ITD mutations are more commonly found in hyperleukocytosis. The outcome of patients with FLT3-ITD mut AML (regardless of concomitant NPM1 mutation) seems to be better in the early phase. However, our study does not allow for drawing conclusions regarding the effectiveness of the LA procedure itself. Nevertheless, our observational findings could be the first step to creating a basis for the ‘fine tuning’ of decision making in hyperleukocytotic AML, since there are no other established parameters for this specific clinical scenario.

Our study indicates that FLT3-ITD mut AML patients with hyperleukocytosis could particularly benefit from LA, or that the hyperleukocytosis in these patients may be less damaging to vital organs, resulting in lower early mortality rates. In a study by Sritana and Auewarakul, blasts of FLT3-mut AML patients showed markedly low or no CD11c expression [34]. This observation could explain the difference in short-term survival that we also see in our study between FLT3-ITD mut and FLT3 wt AML patients presenting with hyperleukocytosis. One could hypothesize that if FLT3-ITD mut blasts are less likely to cause severe leukostasis (e.g., due to less adhesive properties), the LA procedure itself might cause less additional endothelial damage. Furthermore, the hyperproliferative quality of FLT3-ITD mut AML leading to higher average blast percentages in the peripheral blood could allow the apheresis machine to more efficiently remove circulating blasts into the apheresate. Further studies are needed to validate this hypothesis and to better understand the pathophysiology of leukostasis in different subsets of AML defined by genetic features. To use FLT3 mutational status or the most likely associated CD11c-expression as a tool for treatment decisions, rapid diagnostic methods have to be available. As described before, hyperleukocytosis with consecutive leukostasis is a medical emergency. Hence, without point-of-care testing, our hypothesis relating to FLT3 mutational status cannot be incorporated into a treatment algorithm meaningfully. This remains an important limitation of our results and their transferability to a real-world scenario.

LA remains a method for rapid symptom control and is an attempt to avert permanent organ damage. Other predictive parameters are yet to be identified that are rapidly available in the emergency case of leukostasis. Furthermore, there is limited evidence for a survival benefit in symptomatic patients versus asymptomatic patients. Since we do not know from our results whether the LA procedure itself may negatively influence the survival of FLT3 wt patients with hyperleukocytosis or not, routinely performing LA cannot be recommended. Symptomatic leukostasis seems to be the crucial factor indicating severe disease and impending organ failure defining the risk of early death, rather than the method of cytoreduction.

To develop a reliable algorithm that integrates LA into the treatment of symptomatic hyperleukocytotic disease (with somehow proven diagnosis and exclusion of the common differential diagnoses) would have to be studied systemically in a prospective manner in an interventional trial, via the head-to-head comparison of patients undergoing LA versus chemotherapy only in a randomized fashion. So far, no study has been able to clearly answer the question of a benefit from LA, as not only patients with symptomatic leukostasis but also with asymptomatic hyperleukocytosis were included. Theoretically, it is possible that patients without leukostasis may also benefit from LA; the overall role of leukapheresis in the setting of hyperleukocytic AML remains unclear.

## 5. Conclusions

FLT3-ITD mutated AML patients show less early mortality than FLT3 wt patients, although the causal relation remains to be elucidated. If rapid (point-of-care, e.g., in-house real-time polymerase chain reaction) molecular genetic testing is available, the mutational status could guide treatment decisions in the critical early phase of cytoreduction at the time of diagnosis. Since most patients receive more than one LA session, early knowledge of FLT3 mutational status might help in the earlier termination of LA to avoid LA-associated complications (e.g., hemorrhage, electrolyte disturbances) and might allow the development of a more cost-effective approach considering the limited availability and personnel resources. Consequently, we propose the following algorithm, as depicted in Figure 3.

## Figures and Tables

**Figure 1 cancers-16-00058-f001:**
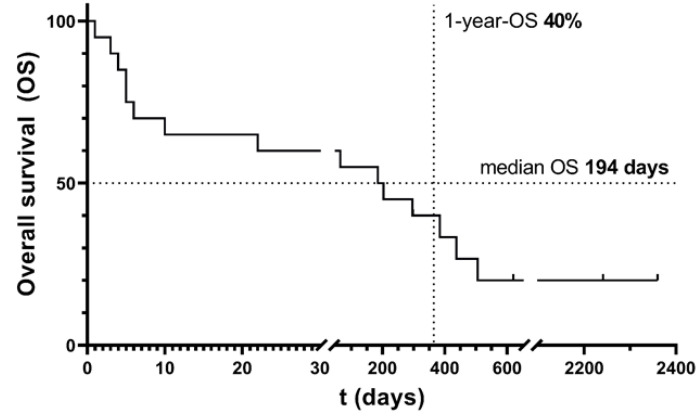
Overall survival of AML patients with hyperleukocytosis undergoing leukapheresis. Notice the different phases: early mortality in the first two weeks, a plateau of leukostasis survivors, a step-by-step decline (late mortality; disease- and therapy-associated), and a second plateau with long-term survivors. N = 20.

**Figure 2 cancers-16-00058-f002:**
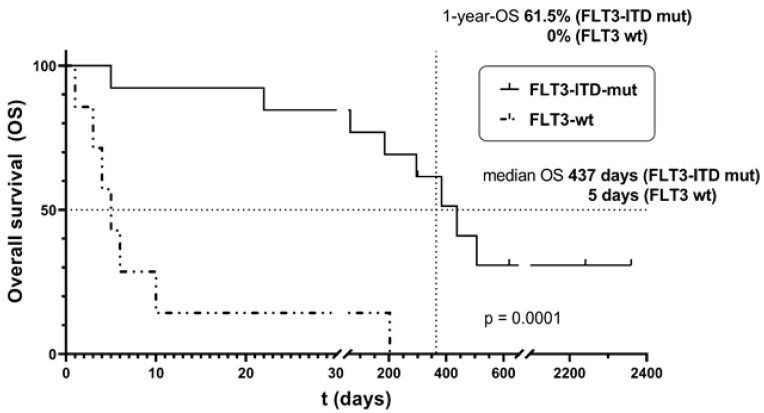
Overall survival of AML patients with hyperleukocytosis undergoing leukapheresis separated by FLT3 mutational status. Almost all patients not surviving the first 14 days after diagnosis had FLT3 wt AML. Log-rank test: *p* = 0.0001.

**Figure 3 cancers-16-00058-f003:**
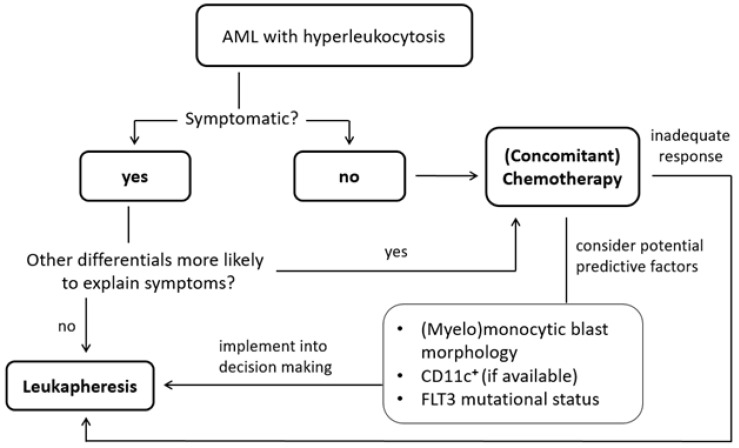
Proposed treatment algorithm for the initial treatment of hyperleukocytic AML. Firstly, and most importantly, the presence of clinical symptoms of leukostasis should be assessed. If symptoms are attributable to probable leukostasis (compatible with the diagnosis of leukostasis according to Novotny’s grading system [35]) without other more likely differential diagnoses, urgent leukapheresis should be considered. In the case of asymptomatic (=non-leukostatic) hyperleukocytosis, and if other differentials are similarly or more likely in symptomatic disease, risk factors (i.a., FLT3 wt status) should be evaluated to further guide decision making. Concomitant cytoreductive chemotherapy is obligatory and must be initiated without delay while the patient is evaluated for risk factors, i.e., (myelo-)monocytic blast morphology, FLT3 status or CD11c expression. As explained, the presence of a FLT3-ITD mutation is ambiguous in its significance—it is a risk factor for hyperleukocytosis but also defines a subset of leukapheresis-treated patients with improved early mortality rate. In patients with risk factors or inadequate response to chemotherapy (with a maximum doses of cytoreductive agents as tolerated/as possible with regard to tumor lysis syndrome), leukapheresis can be considered as an add-on strategy.

**Table 2 cancers-16-00058-t002:** Patient characteristics of leukapheresis cohort.

Patient no.	Age	Sex	Molecular (and cyto)Genetic Features of AML	Symptoms	No. of LA Sessions	Use of Hydroxyurea	Use of Cytarabine	WBC Count Pre-LA (per nL)	WBC Count Post-1st LA (per nL)	WBC Count Reduction	Prothrombin Ratio	Antithrombin III Levels	Serum Creatinine (in mg/dL)	Serum Lactate Dehydrogenase (in U/L)	EASIX Score	Thrombocytes (per nL) pre-LA	Hemoglobin (in g/dL) pre-LA	Survival (days)	Cause of Death
#1	45	m	FLT3-ITD (high ratio)	Visual disturbances	2	no	yes	272.8	134.5	50.7%	80.0%	103.9%	1.00	2400	19.2	125	9.7	296	relapse + sepsis
#2	75	f	Trisomy 8, monosomy 22	Somnolence, anisocoria	3	yes	no	394.1	209.3	46.9%	44.9%	92.5%	4.78	1834	94.3	93	6.9	10	leukostasis
#3	52	m	NPM1, FLT3-ITD (high ratio)	Dyspnea, somnolence	2	yes	yes	194.0	95.3	50.9%	45.2%	n/a	0.89	3388	28.2	107	7.1	384	relapse
#4	59	f	NPM1, FLT3-ITD (high ratio)	Cephalgia	2	no	yes	219.0	82.8	62.2%	48.9%	102.9%	0.48	2874	11.7	118	10.0	185	relapse with leukostasis + steroid-refractory GvHD
#5	52	f	NPM1, FLT3-ITD (high ratio)	DIC, cephalgia	3	yes	yes	191.2	103.8	45.7%	56.5%	111.1%	0.60	574	6.6	52	9.9	506	sepsis after relapse
#6	42	m	FLT3-ITD (high ratio), biallelic CEBPA	Acute kidney failure d/t leukostasis	2	yes	no	213.9	98.2	54.1%	69.4%	n/a	1.78	1883	139.7	24	8.7	2360 (cens.)	*alive at time of analysis*
#7	47	f	NPM1, FLT3-ITD (high ratio)	Dyspnea, severe ischemic leg pain	1	yes	yes	210.7	105.6	49.9%	69.4%	n/a	0.61	1328	23.8	34	8.7	2241 (cens.)	*alive at time of analysis*
#8	66	m	NPM1, FLT3-ITD (high ratio)	Dyspnea	2	yes	yes	249,1	97.9	60.7%	50.0%	78.0%	1.00	658	34.6	19	10.4	65	intestinal perforation d/t neutropenic colitis
#9	56	f	NPM1, FLT3-ITD (low ratio), trisomy 8	Dyspnea	3	no	yes	174.0	153.0	12.1%	62.2%	n/a	0.91	1908	21.2	82	7.9	437	relapse
#10	78	m	FLT3-ITD (high ratio)	Dyspnea, respiratory failure	3	no	yes	586.1	472.1	19.5%	45.1%	n/a	4.38	2817	77.1	160	8.3	620 (lost to f/u)	n/a
#11	64	m	inv(16), FLT3-TKD	Dyspnea, respiratory failure	2	yes	no	231.7	135.3	41.6%	78.9%	n/a	1.16	1332	37.7	41	11.5	5	pneumocystis jirovecii-pneumonia + leukostasis
#12	80	m	FLT3-ITD (low ratio)	Dyspnea, agitation	3	yes	no	321.2	206.5	35.7%	56.3%	n/a	2.03	2109	57.1	75	7.0	22	progressive leukemia
#13	67	m	FLT3 wt	Respiratory failure	2	no	yes	396.6	310.9	21.6%	31.2%	n/a	5.77	2652	259.4	59	6.4	1	septic shock + leukostasis
#14	79	m	NPM1, ASXL1, TET2	Dyspnea, respiratory failure	3	no	yes	192.7	97.5	49.4%	49.6%	n/a	2.26	1316	52.2	57	9.8	6	pneumonia
#15	75	f	NPM1, trisomy 8	Paresthesia, visual disturbances, vertigo, DIC	1	yes	yes	147.6	72.8	50.7%	36.8%	83.5%	3.43	2104	97.5	74	8.2	3	sepsis after relapse
#16	62	f	NPM1, FLT3-ITD (high ratio)	Dyspnea, somnolence	3	no	yes	180.1	95.1	47.2%	44.9%	n/a	0.86	3025	63.5	41	7.8	5	cerebral hemorrhage
#17	39	f	NPM1, FLT3-ITD	Dyspnea, tinnitus	2	no	yes	115.3	69.2	40.0%	69.4%	102.3%	0.65	1144	20.7	36	9.9	358 (cens.)	*alive at time of analysis*
#18	62	f	RUNX1, TET2	Dyspnea, respiratory failure	1	yes	yes	170.2	113.6	33.3%	48.7%	69.5%	3.35	561	48.2	39	7.8	349	relapse
#19	45	f	NPM1, FLT3-ITD	Visual disturbances, cephalgia	1	yes	yes	117.7	71.8	39.0%	61.9%	94.8%	0.73	1388	21.6	47	10.4	356 (cens.)	*alive at time of analysis*
#20	18	m	t(10;11)	Dyspnea, respiratory failure	1	no	yes	146.2	73.0	50.1%	33.3%	51.5%	2.81	1146	29.5	109	5.8	4	leukostasis

ASXL1: additional sex combs-like 1; CEBPA: CCAAT/enhancer-binding protein alpha; cens: censored; d/t: due to; DIC: disseminated intravascular coagulation; EASIX: endothelial activation and stress index; f: female; f/u: follow-up; GvHD: graft-versus-host disease; HU: hydroxyurea; ITD: internal tandem duplication; LA: leukapheresis; m: male; n/a: not applicable or available; NPM1: nucleophosmin 1; OS: overall survival; RUNX1: runt-related transcription factor 1; WBC: white blood cell; wt: wild type. High (allelic) ratio for FLT3-ITD was defined as a FLT3-ITD to FLT3 wt ratio of ≥0.5.

**Table 3 cancers-16-00058-t003:** Summarized patient characteristics of leukapheresis cohort.

Characteristics	Leukapheresis Cohort (*n* = 20)	*p* Value
**Age (years)**	**58 (18–80)**	
□ in FLT3-ITD mut AML □ in FLT3 wt AML	62.9 (39–80) 55.7 (18–79)	(ITD vs. wt) = 0.3529
**Average leukapheresis sessions/patient**	**2.1 (1–3)**	
**Non-favorable risk AML**	**17 (85%)**	
□ FLT3-ITD mut AML	13 (65%)	
**Clinical symptoms of leukostasis**	**20 (100%)**	
□ respiratory *(requiring oxygen supply)*	12 (60%)	
□ neurological	10 (50%)	
**Cytoreductive agent**	**20 (100%)**	
□ hydroxyurea (HU)	13 (65%)	
□ cytarabine	17 (85%)	
□ HU + cytarabine	7 (35%)	
**Average leukocyte count pre-treatment (per nanoliter)**	**236.2 (115.3–586.1)**	
□ in FLT3-ITD mut AML □ in FLT3-wt AML	199.1 (51.6–586.1) 205.3 (86.4–396.6)	(ITD vs. wt) = 0.8679
**Leukocyte count after leukapheresis treatment (per nanoliter)**	**123.7 (14.7–472.1)**	
□ after 1st leukapheresis □ after last leukapheresis	139.9 (69.2–472.1) 92.1 (47.5–288.7)	
**Relative leukocyte count reduction (percent)**	**41.5 (4.0–71.6)**	
□ after 1st leukapheresis □ after last leukapheresis	43.1 (12.1–62.2) 57.8 (27.2–92.4)	
**EASIX pre-treatment**	**57.2 (6.6–259.4)**	
□ in FLT3-ITD mut AML	88.4 (29.5–259.4) 40.4 (6.6–139.7)	(ITD vs. wt) = 0.2767
□ in FLT3 wt AML		
**Median OS (days)**	**194** (1–2379)	
□ in FLT3-ITD mut AML □ in FLT3 wt AML	437 (5–2379) 5 (1–203)	(ITD vs. wt) = **0.0001**
**14-day survival (percent)**	**65**	
□ in FLT3-ITD mut AML □ in FLT3 wt AML	92.3 14.3	

Values are presented as mean (range). AML: acute myeloid leukemia; EASIX: endothelial activation and stress index; FLT3: Fms related receptor tyrosine kinase 3; HU: hydroxyurea; ITD: internal tandem duplication; mut: mutated; OS: overall survival; wt: wild type.

**Table 4 cancers-16-00058-t004:** Patient characteristics of non-leukapheresis cohort.

Patient no.	Age	Sex	Molecular (and cyto)Genetic Features of AML	Symptoms	Use of Hydroxyurea	Use of Cytarabine	WBC Count Pre-Treatment (per nL)	WBC Count Post 48 h of Treatment (per nL)	WBC Count Reduction	Prothrombin Ratio	Antithrombin III Levels	Serum Creatinine (in mg/dL)	Serum Lactate Dehydrogenase (in U/L)	EASIX Score	Thrombocytes (per nL) pre-Treatment	Hemoglobin (in g/dL) pre-Treatment	Survival (days)	Cause of Death
#1	80	m	RUNX1::RUNX1T1	Respiratory failure	no	yes	120.4	97.0	19.4	56%	62%	1.34	1100	43.4	34	7.4	11	Leukostasis
#2	72	m	NPM1	No clinical symptoms	yes	no	140.7	86.9	38.2	66%	n/a	0.88	1000	4.8	183	7.3	83 (lost to f/u)	n/a
#3	45	f	NPM1, FLT3-ITD (low ratio)	No clinical symptoms	yes	no	219.8	80.7	63.3	47%	n/a	0.85	1252	35.5	30	8.5	1950	Diffuse alveolar hemorrhage + sepsis + severe gastrointestinal GvHD
#4	68	m	SF3B1	No clinical symptoms	yes	no	148.4	119.3	19.6	75%	n/a	1.22	3071	44.6	84	8.8	510 (cens.)	**Alive at time of analysis**
#5	48	m	NPM1, FLT3-ITD (high ratio)	No clinical symptoms	yes	yes	109.5	81.51	25.6	75%	105,7	1.05	1257	13.5	98	14.0	902 (cens.)	**Alive at time of analysis**
#6	64	m	FLT3-ITD, CEBPA	Dyspnea	yes	yes	226.3	208.6	7.8	54%	n/a	0.90	1446	5.6	234	11.2	566	Relapse + gastrointestinal hemorrhage
#7	44	f	TET2, WT1, KRAS	No clinical symptoms	yes	no	161.6	n/a	n/a	111%	n/a	0.58	1894	5.9	187	6.0	412	Relapse
#8	76	f	NRAS, TET2, KIT	Respiratory failure	yes	yes	409.0	287.9	29.6	76%	96%	2.29	1433	88.7	37	5.5	197	Relapse
#9	30	f	complex aberrant karyotype	No clinical symptoms	(only prednisone)		170.5	101.0	40.8	77%	88%	0.55	1694	21.7	43	7.5	2865 (cens.)	**Alive at time of analysis**
#10	53	f	trisomy 13, IDH1	No clinical symptoms	no	yes	104.6	130.5	−24.8	88%	n/a	0.83	480	11.1	36	9.6	582 (cens.)	**Alive at time of analysis**

CEBPA: CCAAT/enhancer-binding protein alpha; EASIX: endothelial activation and stress index; f: female; f/u: follow-up; GvHD: graft-versus-host disease; HU: hydroxyurea; IDH1: isocitrate dehydrogenase 1; ITD: internal tandem duplication; m: male; n/a: not applicable or available; NPM1: nucleophosmin 1; NRAS: neuroblastoma ras viral oncogene homolog; OS: overall survival; RUNX1: runt-related transcription factor 1; SF3B1: splicing factor 3b subunit 1; TET2: tet methylcytosine dioxygenase 2; WBC: white blood cell; wt: wild type; WT1: Wilms’ tumor gene 1. High (allelic) ratio for FLT3-ITD was defined as a FLT3-ITD to FLT3 wt ratio of ≥0.5.

**Table 5 cancers-16-00058-t005:** Summarized patient characteristics of non-leukapheresis cohort.

Characteristics	Non-Leukapheresis Cohort (*n* = 10)	*p* Value *(Compared to LA Cohort)*
**Age (years)**	**58 (30–80)**	0.5390
**Non-favorable risk AML**	**7 (70%)**	
□ FLT3-ITD mut AML	3 (30%)	
**Clinical symptoms of leukostasis**	**3 (30%)**	
□ respiratory *(requiring oxygen supply)*	3 (100%)	
□ neurological	none	
**Cytoreductive agent**	**10 (100%)**	
□ hydroxyurea (HU)	5 (50%)	
□ cytarabine	6 (60%)	
□ HU + cytarabine	2 (20%)	
□ other (e.g., prednisone)	1 (10%)	
**Average leukocyte count pre-treatment (per nanoliter)**	**181.1 (104.6–409.0)**	0.5965
□ in FLT3-ITD mut AML □ in FLT3 wt AML	179.3 (104.6–409.0) 185.2 (109.5–226.3)	(ITD vs. wt) = 0.9312
**Leukocyte count after 48 h of cytoreductive treatment (per nanoliter)**	**132.6 (80.9–287.9)**	
**Relative leukocyte count reduction (percent)**	**24.4 (−24.8–63.9)**	0.0096
**EASIX pre-treatment** □ in FLT3-ITD mut AML □ in FLT3 wt AML	**27.5 (4.8–88.7)** 18.2 (5.6–35.5) 31.4 (4.8–88.7)	0.1390
		0.9999 0.1367 (ITD vs. wt) = 0.9999
**Median OS (days)**	**412 (11–2865)**	
□ in FLT3-ITD mut AML □ in FLT3 wt AML	1258 (566–1950) 197 (11–2865)	(ITD vs. wt) = 0.1088
**14-day survival (percent)**	**90**	

Values are shown as mean (range). AML: acute myeloid leukemia; EASIX: endothelial activation and stress index; FLT3: Fms related receptor tyrosine kinase 3; HU: hydroxyurea; ITD: internal tandem duplication; LA: leukapheresis; mut: mutated; OS: overall survival; wt: wild type.

**Table 6 cancers-16-00058-t006:** Uni- and multivariate analysis of parameters influencing 14-day survival.

	14-Day Survival		
Variable	*p* Value *(Univariate Analysis)*	Odds Ratio (CI 95%; *Multivariate Analysis*)	*p* Value *(Multivariate Analysis)*
Age (≥60 vs. <60 years)	0.057	0.12 (0.002–2.17)	0.1900
FLT3-ITD (mut vs. wt)	**0.007**	19.02 (1.48–682.6)	**0.0424**
Relative WBC count reduction ≥50% after first leukapheresis (yes vs. no)	1.000	0.85 (0.024–27.90)	0.9169

CI: confidence interval; FLT3: Fms related receptor tyrosine kinase 3; ITD: internal tandem duplication; mut: mutated; WBC: white blood cell; wt: wild type.

## Data Availability

The data generated in the present study are included in the figures and tables of this article.

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
