# Peer review of "Symptomatic Patients with Hyperleukocytic FLT3-ITD Mutated Acute Myeloid Leukemia Might Benefit from Leukapheresis"

_cancers, 2023, doi:10.3390/cancers16010058_

Round 1

Reviewer 1 Report

Comments and Suggestions for Authors

This reviewer was in charge of therapeutic apheresis at a large teaching hospital for over 15 years and therefore was much interested in reading about another center's experience. I found the in formation presented in a confusing manner which seemed to be slanted to make the authors' point and did not convey the facts in an objective manner.

My impression is that they need to answer some of the questions I pose below to convince me that their findings were correctly interpreted.

1. It is NOT acceptable to this reader to have the data (Suppl Table 2) in the supplementary data. This prevents the reader from making actual comparisons of the differences between the leukapheresis (LA) group and the no-LA group.

2. Suppl Table 2 raises many questions. 

First, are patients 1 and 2 included in the survival curve?  The authors state that patients 1 and 2 (out of 10 total, 20%) were "lost to followup". How can patient 1 with respiratory failure be "lost to followup"? Patient 1 is reported as having a 6 day survival but he was lost to followup. Patient 2 had an 83 day followup. If these two are included in the Kaplan Meier analysis, this significantly shortens the survival. Please clarify if these patients were included in the Kaplan Meier analysis. IF they were please recalculate without including their survival.

2. Personally I would have expected for leukapheresis (LA) to prolong the survival of the patients in the short term since FLT3 mutated patients with high alleleic ratios are usually very ill at presentation due to rapid rise in WBC, as were the 20 LA patients. Table 2 shows that 9 of 20 had high FLT3 allelic ratios. It is not expected that LA at diagnosis would prolong life beyond the immediate period post diagnosis, specfically in the patients who were FLT3 mutated who have the worst prognosis.

3. To explain their findings, the authors need to present the data somewhat differently. First of all, the authors need to explain why they focus on survival "UP TO ONE YEAR" (abstract). When we compare the LA group to the Non-LA group, we see that the LA group had 4 patients (of 20) who were survivors to the time point of when the analysis was done, as compared to 4 of only 10 of the Non-LA group. It appears that the LONG term survival was actually longer in the NON-LA group. 

4. Furthermore in Figure 2, the authors separate the Kaplan Meier curve by FLT3 mutation status whereas most authorities indicate that only those with high allelic burden are in danger and that low alleleic ratio should not be considered as such an adverse prognosis. What happens if the authors redo this analysis of hyperleukocytosis patients undergoing LA by FLT3 status but including the ONLY the FLT3 high allelic ratio versus the low allelic ratio patients together with the FLT3 negative patients?

5. LA is a procedure which can provoke tumor lysis, bleeding and other events and therefore should not be used in patients who are not symptomatic sincce chemotherapy can lower WBC within days. Since the clinical decision for performing LA is made usually in the emergency dept, there is no way to know in the first hours after presentation whether or not the patient is FLT3 positive (or the allelic ratio). Therefore when one thinks objectively about the data, it is hard to find an "actionable" lesson here. We have to decide on LA according to the symptoms and we can not wait for the FLT3 mutation status to decide on LA. In the long run, the authors are possibly making a case for LA in symptomatic patients, which I completely agree with  despite the fact that to me, the FLT3 status was not actually of much importance here.

6. The authors do not give us any information about the post induction therapy these patients had. Were the LA and non_LA patients treated in an identifal manner? Obviously the long term prognosis is more dependent on the intensity of and the response to, long term therapy, whatever that may have been.

7. Minor point: AML is a very heterogeneous disease as can be seen in the tables. Thus the comparison of these two groups which are so small and not of equal size seems problematic.

Author Response

Dear Editor-in-Chief:,

Thank you for giving us the opportunity to submit a revised draft of our manuscript.

We appreciate the time and effort that you and the reviewers have dedicated to providing your valuable feedback on our manuscript. In the new version of our manuscript, we were able to incorporate substantial changes to reflect the suggestions provided by the reviewers, and we have highlighted the changes within the manuscript.

Here is a point-by-point response to the reviewers’ comments and concerns.

---

  1. It is NOT acceptable to this reader to have the data (Suppl Table 2) in the supplementary data. This prevents the reader from making actual comparisons of the differences between the leukapheresis (LA) group and the no-LA group.

Thank you for raising this important point. In the new version of our manuscript, we included all patients’ data into the main manuscript.

  1. Suppl Table 2 raises many questions. 

First, are patients 1 and 2 included in the survival curve?  The authors state that patients 1 and 2 (out of 10 total, 20%) were "lost to follow-up". How can patient 1 with respiratory failure be "lost to follow-up"? Patient 1 is reported as having a 6 day survival but he was lost to follow-up. Patient 2 had an 83 day follow-up. If these two are included in the Kaplan Meier analysis, this significantly shortens the survival. Please clarify if these patients were included in the Kaplan Meier analysis. IF they were please recalculate without including their survival.

Patient #1 was transferred to a palliative care unit outside of our facility on his own request and therefore was considered ‚lost to follow-up‘. Taking your point very seriously, we now contacted the palliative care unit and could obtain the following information: patient #1 died on day 11. We recalculated median OS of non-leukapheresis treated patients accordingly. Patient #2 was a real rare case of ‚loss to follow-up‘ – we reviewed the patient’s chart at our facility and called the external hospital he was transferred to and where the patient last presented in; unfortunately there are no available records of any visits beyond day 83. Therefore, the survival data for patient #2 were censored.

  1. Personally I would have expected for leukapheresis (LA) to prolong the survival of the patients in the short term since FLT3 mutated patients with high allelic ratios are usually very ill at presentation due to rapid rise in WBC, as were the 20 LA patients. Table 2 shows that 9 of 20 had high FLT3 allelic ratios. It is not expected that LA at diagnosis would prolong life beyond the immediate period post diagnosis, specifically in the patients who were FLT3 mutated who have the worst prognosis.

This is indeed a very important point, where we fully agree. The main issue here is that prospective data is not available for this exact question/comparison (long term survival with initial leukapheresis vs. without leukapheresis). We also believe that leukapheresis might be useful/might impact the survival only in the initial phase of combatting leukostasis-induced organ dysfunction. Long term prognosis was mainly influenced by disease progression or therapy-related complications. We have incorporated these thoughts into the text of our new version of the manuscript (lines 272 f., 368 f.).

  1. To explain their findings, the authors need to present the data somewhat differently. First of all, the authors need to explain why they focus on survival "UP TO ONE YEAR" (abstract). When we compare the LA group to the Non-LA group, we see that the LA group had 4 patients (of 20) who were survivors to the time point of when the analysis was done, as compared to 4 of only 10 of the Non-LA group. It appears that the LONG term survival was actually longer in the NON-LA group. 

Our focus was early mortality (i.e. 14-day-mortality in our study). We have incorporated your suggestion in the abstract and throughout the manuscript to highlight that our main investigational goal was to study differences in the very early critical phase immediately after diagnosis.  

  1. Furthermore in Figure 2, the authors separate the Kaplan Meier curve by FLT3 mutation status whereas most authorities indicate that only those with high allelic burden are in danger and that low allelic ratio should not be considered as such an adverse prognosis. What happens if the authors redo this analysis of hyperleukocytosis patients undergoing LA by FLT3 status but including the ONLY the FLT3 high allelic ratio versus the low allelic ratio patients together with the FLT3 negative patients?

Thank you for this important suggestion. It would have been interesting to explore this aspect. However, in the case of our study,   the newer classification system (WHO 2022) does not differentiate any longer between the subclasses of high and low ratios for FLT3-ITD mutated AML, since the prognostic impact of allelic burden is not any longer considered as high/relevant as previously thought.

Moreover, the vast majority of our patients treated with leukapheresis had anyway an AML with high ratio FLT3-ITD mutation (9 out of 13 FLT3-positive patients, and 2 patients in whom the allelic ratio was not calculated). If we apply the suggested changes: 1 of 2 FLT3-ITD low ratio-patients surviving only 22 days would impact OS only slightly, but not early (14-day-) mortality, and the patients with unknown ratio cannot be clearly assigned to either group. Thus, it would most likely not be feasible to separate the subgroups in the suggested way and most likely not make a difference that would change the direction of our results.

  1. LA is a procedure which can provoke tumor lysis, bleeding and other events and therefore should not be used in patients who are not symptomatic since chemotherapy can lower WBC within days. Since the clinical decision for performing LA is made usually in the emergency dept, there is no way to know in the first hours after presentation whether or not the patient is FLT3 positive (or the allelic ratio). Therefore, when one thinks objectively about the data, it is hard to find an "actionable" lesson here. We have to decide on LA according to the symptoms and we cannot wait for the FLT3 mutation status to decide on LA. In the long run, the authors are possibly making a case for LA in symptomatic patients, which I completely agree with despite the fact that to me, the FLT3 status was not actually of much importance here.

We thank the reviewer for bringing up this issue. Leukapheresis surely can lead to life-threatening complications and we agree on avoiding ‚prophylactic‘ LA in asymptomatic patients. However, at our facility, the turnover time for FLT3 PCR is only ~24 hours. Since most patients with hyperleukocytosis need ≥1 leukapheresis session (as seen in our cohort, each one day apart), knowledge about mutational status could avoid repeated sessions and therefore potential LA-related complications with each procedure. Furthermore, the possibility of better decision making (FLT3-guided) might allow a more cost-effective approach considering the limited availability and personnel resources. We acknowledge that the mutational status at the timepoint of diagnosis is not a ‚revolutionary game changer‘, but most certainly could act as an additional tool of guidance in borderline situations, where the risk-benefit-ratio is doubtful to confidently determine from a clinical point of view. We obviously also could not base our (clinical) decisions on FLT3 status since this was a retrospective finding and realization – the validation of our hypothesis would need prospective investigation.

  1. The authors do not give us any information about the post induction therapy these patients had. Were the LA and non_LA patients treated in an identifal manner? Obviously the long term prognosis is more dependent on the intensity of and the response to, long term therapy, whatever that may have been.

Thank you for making this point. We agree that long term prognosis is highly dependent on the intensity of treatment and consolidation strategies. However, our focus was early mortality, which is significantly influenced by organ-damaging leukostasis. Following our institutions standard operating procedures (SOPs), all patients eligible for allogeneic transplantation (if other than favorable-risk AML) were treated with intensive chemotherapy or ‚non-intensive‘ induction strategies followed by transplant, provided that a treatment response was achieved. Long term outcomes were influenced by relapse and transplant-associated complications.

  1. Minor point: AML is a very heterogeneous disease as can be seen in the tables. Thus the comparison of these two groups which are so small and not of equal size seems problematic.

The reviewer has raised an important point here. However, the sample size of our cohort may be small, but nonetheless comparable to other published reports. This very specific subset of patients is a difficult-to-study population, due to several reasons which we discussed in the manuscript, being aware of the limitations and in a self-reflective manner. According to the WHO 2022 classification, we have disclosed the different genetic subtypes. In addition, the smaller sample size of the non-leukapheresis group is mainly due to the paucity of patients presenting with marked hyperleukocytosis but without symptoms, in whom we would not perform leukapheresis.

---

By this, we sincerely hope that we have properly addressed all points raised by both reviewers and that our manuscript will be acceptable in its present version for publication in your prestigious journal.

On behalf of all authors, I am remaining

Sincerely Yours,

Kiavasch Farid

Reviewer 2 Report

Comments and Suggestions for Authors

Farid et. al described their unicentric retrospective experience on a controversial issue in the field of AML: the use of leukapheresis for AML cases with symptomatic hyperleukocytosis.

Their sample size is not large but comparable to other published reports, and the results described are interesting in that the presence of mutations in the FLT3 gene would appear to have a protective effect on leukostasis mortality for patients undergoing leukapheresis.

Just a few questions:

-        In methods, I suggest adding a description of the enrolment of the control group: for example, is the time period the same?

-        Please correct the row of patient number 20 in table 2: I am afraid there is an error in the number of white blood cells.

-        In methods, specify that the enrolment concerned patients with symptomatic hyperleukocytosis, as described in lines 251-252 of the discussion. And by the way, if this was the case, why is it stated in the description of the results (lines 135-136) that only 95% had symptoms related to hyperleukocytosis? please verify these data.

-        I suggest talking about 14-day mortality in univariate and multivariate logistic regression analysis and then change the odds ratios.

-        The authors should formulate biological hypotheses in the discussion of why leukopheresis should work better in flt3 mutated AMLs.

-        In the discussion, the role of CD11c antigen expression on leukaemic blasts as an agent of hyperviscosity is mentioned; what about its expression in your population? Are these data available?

-        Probably the proposal of a therapeutic algorithm for approaching patients with hyperleukocytosis is not adequate given the retrospective nature of the study based on a very small sample size.

Author Response

Dear Editor-in-Chief:,

Thank you for giving us the opportunity to submit a revised draft of our manuscript.

We appreciate the time and effort that you and the reviewers have dedicated to providing your valuable feedback on our manuscript. In the new version of our manuscript, we were able to incorporate substantial changes to reflect the suggestions provided by the reviewers, and we have highlighted the changes within the manuscript.

Here is a point-by-point response to the reviewers’ comments and concerns.

---

-        In methods, I suggest adding a description of the enrolment of the control group: for example, is the time period the same?

We thank the reviewer for pointing this out. The time period of enrolment for the control group is Sep 2015 – Oct 2022 and thus, similar to the LA treated patients. We added this to the methods section.

-        Please correct the row of patient number 20 in table 2: I am afraid there is an error in the number of white blood cells.

We thank the reviewer for giving us this remark. The WBC count pre-leukapheresis was 146.2/nl, not 14.2/nl.

-        In methods, specify that the enrolment concerned patients with symptomatic hyperleukocytosis, as described in lines 251-252 of the discussion. And by the way, if this was the case, why is it stated in the description of the results (lines 135-136) that only 95% had symptoms related to hyperleukocytosis? please verify these data.

Thank you for highlighting this inconsistency in our previous version of the manuscript. As described in lines 296 ff., all patients had symptomatic disease. The presence and type of symptoms are shown in Table II in detail. One of the patients (i.e. patient #5) showed DIC at presentation (alongside some cephalgia and exertional dyspnea) and we had not attributed her symptoms to possible leukostasis when we first performed the analysis of our data. Repeated study of the patient’s chart brought us to the conclusion, that the decision to perform leukapheresis was based on the presence of symptomatic disease (like every other patient in our cohort). Thus, we had revised the lines in the discussion, but forgot to do so in the results section. We changed the description of the results accordingly.

-        I suggest talking about 14-day mortality in univariate and multivariate logistic regression analysis and then change the odds ratios.

We thank the reviewer for this suggestion. We incorporated this thought into Table VI in the new version of the manuscript.

-        The authors should formulate biological hypotheses in the discussion of why leukopheresis should work better in flt3 mutated AMLs.

Since we did not perform any laboratory studies on the samples of our patients, the following hypotheses – as requested – will be very speculative. One could hypothesize that if FLT3-ITD mutated blasts are less likely to cause severe leukostasis e.g. due to less adhesive properties, the leukapheresis procedure itself might cause less additional endothelial damage. Also, the hyperproliferative quality of FLT3-ITD mutated AML leading to higher average blast percentages in the peripheral blood allows the apheresis machine to more efficiently remove circulating blasts into the apheresis bag. We added these hypotheses to the discussion. These considerations have been incorporated into the new version of our manuscript on page 12.

-        In the discussion, the role of CD11c antigen expression on leukaemic blasts as an agent of hyperviscosity is mentioned; what about its expression in your population? Are these data available?

Unfortunately, we do not have this specific piece of information. Indeed, it would have been interesting to have these data. Unfortunately, our routine flow cytometry panel did not include CD11c at the time of the diagnostic workup of the patients.  We are planning to include CD11c in our standard panel in the future.

-      Probably the proposal of a therapeutic algorithm for approaching patients with hyperleukocytosis is not adequate given the retrospective nature of the study based on a very small sample size.

The sample size of our cohort may be small, but nonetheless comparable to other published reports, as you already had mentioned. This very specific subset of patients is a difficult-to-study population, due to several reasons discussed, aware of the limitations and in a self-reflective manner. The algorithm is not solely derived from our studied population, but also from the literature review we conducted. The proposed algorithm summarizes the pre-existing evidence/suggestions and additionally integrates the new finding of the potential relevance of FLT3 mutational status. It serves as a guidance for the clinician rather than a strict set of rules.

---

By this, we sincerely hope that we have properly addressed all points raised by both reviewers and that our manuscript will be acceptable in its present version for publication in your prestigious journal.

On behalf of all authors, I am remaining

Sincerely Yours,

Kiavasch Farid

Reviewer 3 Report

Comments and Suggestions for Authors

The benefit and effectiveness of leukapheresis (LA) in the initial treatment of acute myeloid leukemia (AML) patients presenting with leukostasis in hyperleukocytosis remains controversial in published literature. In this retrospective analysis, Farid et al. report on the role of LA in 20 consecutive patients with AML with leukostasis in hyperleukocytosis.  They succeeded in identifying subsets of patients who could benefit from emergency LA (FLT3-ITD mut). 

Additionally, the authors composed a management algorithm for AML patients, that could be applied in routine practice. This article is worth publishing regardless of its retrospective nature and small patient sample size. 

Author Response

Dear Editor-in-Chief:,

Thank you for giving us the opportunity to submit a revised draft of our manuscript.

We appreciate the time and effort that you and the reviewers have dedicated to providing your valuable feedback on our manuscript. We are very much thankful to this reviewer for the positive feedback!

By this, we sincerely hope that our manuscript will be acceptable in its present version for publication in your prestigious journal.

On behalf of all authors, I am remaining

Sincerely Yours,

Kiavasch Farid

Round 2

Reviewer 1 Report

Comments and Suggestions for Authors

Abstract: The authors write:

showed a better outcome concerning early mortality after emergency LA”. The word “concerning” is not clear in this context.  Better to say “showed better short term outcome with lower early mortality after emergency LA.” This is 12 words instead of 10.

I have no other suggestions for revisions.

Author Response

Thank you once again for giving us the opportunity to submit a revised draft of our manuscript.

---

Thank you - we changed the phrase in the abstract accordingly.

---

On behalf of all authors, I am remaining

Sincerely Yours,

Kiavasch Farid

Reviewer 2 Report

Comments and Suggestions for Authors

Thank you!

Author Response

Thank you once again for giving us the opportunity to submit a revised draft of our manuscript.

---

We would like to thank you for your insightful comments and suggestions.

---

On behalf of all authors, I am remaining

Sincerely Yours,

Kiavasch Farid